



# Scaled distribution mapping: a bias correction method that preserves raw climate model projected changes

Matthew B. Switanek[1], Peter A. Troch[2], Christopher L. Castro[2], Armin Leuprecht[1], Hsin-I Chang[2], Rajarshi Mukherjee[2], Eleonora M.C. Demaria[3]

[1]Wegener Center for Climate and Global Change, University of Graz, Graz, 8010, Austria
[2]Department of Hydrology and Atmospheric Sciences, University of Arizona, Tucson, 85721, USA
[3]Southwest Watershed Research Center, USDA – Agricultural Research Service, Tucson, 85719, USA

*Correspondence to*: Matthew B. Switanek (matthew.switanek@uni-graz.at)

**Abstract.** Commonly used bias correction methods such as quantile mapping (QM) assume the function of error correction values between modelled and observed distributions are stationary or time-invariant. This article finds that this function of the error correction values cannot be assumed to be stationary. As a result, QM lacks justification to inflate/deflate various moments of the climate change signal. Previous adaptations of QM, most notably quantile delta mapping (QDM), have been developed that do not rely on this assumption of stationarity. Here, we outline a methodology called scaled distribution mapping (SDM), which is conceptually similar to QDM, but more explicitly accounts for the frequency of rain days and the likelihood of individual events. The SDM method is found to outperform QM, QDM and detrended QM in its ability to better preserve raw climate model projected changes to meteorological variables such as temperature and precipitation.

**Keywords:** bias correction, stationarity, quantile mapping, precipitation, temperature, climate change signal

## 1 Introduction

Bias correction of climate model projections is essential in order to properly assess the impacts of climate change on human and environmental resources (Berg et al., 2003; Ines and Hansen, 2006; Muerth et al., 2013; Teng et al., 2015). Removing model bias is especially important for impact studies involving hydrological models, where runoff is a nonlinear function of precipitation (Christensen et al., 2008; Mauer and Hidalgo, 2008; Hagemann et al., 2011). Global climate models (GCMs) provide large-scale projections for many climate variables (IPCC, 2013). However, many climate processes and landscape features are not resolved at the coarse resolution of current GCMs. To bridge this gap, regional climate models (RCMs) are commonly used to downscale GCM data to a higher resolution. Even though RCMs can provide added value (Fowler et al., 2007; Feser et al., 2011; Di Luca et al., 2013; Kotlarski, 2014), systematic errors in the model output still exist (Mearns et al., 2012; Sillmann et al., 2013).

Numerous statistical bias correction methodologies have been developed to remove systematic model errors (Schmidli et al, 2006; Boé et al., 2007; Lenderink et al., 2007; Leander et al., 2008; Gellens and Roulin, 2012; Chen et al.,





2013). The methods adjust the modelled mean, variance, and/or higher moments of the distribution of climate variables, to more closely match the observations. Quantile mapping (QM) has been a widely used method due to its ability to handle higher order moments in addition to being computationally efficient (Wood et al., 2004; Piani et al., 2010; Themeßl et al., 2011; Gudmundsson et al., 2012; Teutschbein and Seibert, 2013). Standard QM assumes that the function of error correction

values found in a calibration period can be applied to any time period of interest. This is referred to as the stationarity assumption or the time-invariant assumption (Christensen et al., 2008; Maraun, 2012; Themeßl et al., 2012; Brekke et al., 2013; Chen et al., 2013). The assumption of stationarity, in QM, is responsible for inflating (or altering) the raw model projections of climate change (Maruer and Pierce, 2014).

In this study, we focus specifically on the performance of bias correction and not issues related to

downscaling/upscaling (Mauran, 2013). We have divided the study in two main sections: Sect. 3 and Sect. 4. In Sect. 3, first we begin by testing the stationarity assumption used in QM and the implications of this assumption. Second, we use a synthetic example to investigate the potential advantages of using a parametric instead of a non-parametric approach. Third, we illustrate the problems associated with validating bias correction methods using a split-sample or cross-validation test. In Sect. 4, a new bias correction method called scaled distribution mapping (SDM) is outlined. Finally, Sect. 5 compares and

discusses the performances of SDM with other methods.

## 2 Data

Climate model data in this study uses projections of daily mean temperature and precipitation values from the KNMI-RACMO22E regional climate model. The KNMI-RACMO22E RCM was forced with the ICHEC-EC-EARTH GCM, and it is one of the model projection runs in the EURO-CORDEX project. The data can be found on any of the ESGF repositories containing

EURO-CORDEX (e.g., https://pcmdi.llnl.gov/projects/esgf-llnl/). The model data for the years 1951-2005 and 2006-2100 correspond to the historical and RCP 8.5 (ESGF naming: r1i1p1) scenarios, respectively. Observational data for mean temperature and precipitation were obtained from the E-OBS data set (Haylock et al., 2008). The KNMI-RACMO22E climate data was upscaled from its original 0.11° resolution to the 0.5° E-OBS resolution.

## 3 Bias correction: Methods, limitations and evaluation

Over the years, numerous bias correction methods have been developed using univarite and multivariate approaches (Gellens and Roulin, 1998; Wood et al., 2004; Schmidli et al., 2006; Boe et al., 2007; Leander and Buishand, 2007; Lenderink et al., 2007; Li et al., 2010; Maraun et al., 2010; Piani et al., 2010; Themeßl et al., 2011; Piani and Haerter, 2012). In this study, we focus our study on univariate bias correction methods.

Many popular existing bias correction methods have been reviewed and compared and quantile mapping (QM) was

found to outperform other methods (Gudmundsson et al., 2012; Teutschbein and Seibert, 2012; Teutschbein and Seibert



2013; Chen et al., 2013). At the same time, studies have pointed out serious problems that arise when using QM for bias correction (Hagemann et al., 2011; Themeßl et al., 2012). In particular, the method can alter the raw model projected changes (Themeßl et al., 2012; Maurer and Pierce, 2014). This inflation or deflation of the raw simulated climate change signal exists as an artefact of the stationarity assumption. The impact that the stationarity assumption has on QM bias corrected data is
discussed in more detail in Sect. 3.1.

More recently, the standard non-parametric QM method has been adapted to more explicitly preserve the raw modelled climate change signals (Michelangeli et al., 2009; Olsson et al., 2009; Willems and Vrac, 2011; Sunyer et al., 2014; Wang and Chen, 2014; Cannon et al., 2015). Hempel et al. (2013) and Bürger et al. (2013) both used a form of detrended QM (DETQM) that better preserved monthly trends, but the daily values still are subject to the stationarity assumption which
can ultimately result in altering the raw modelled projected change. Extending previous work (e.g., Olsson et al., 2009; Bürger et al., 2013), Cannon et al. (2015) modified the QM method and outlined an approach called quantile delta mapping (QDM). QDM is a break from other typical QM methods insofar that it is not constrained by the stationarity assumption. An example of QM versus QDM is shown in Figure 1. In the traditional QM method, a raw modelled value is always corrected by the same value of bias or error that is determined by its respective quantile in the calibration period. On the other hand,
QDM multiplies observed values by the ratio of the modelled values (period of interest divided by calibration period) at the same quantiles. Our proposed bias correction methodology, scaled distribution mapping (SDM), share some similarities with QDM, however there are three important distinctions: 1) SDM uses a parametric model instead of a non-parametric one, 2) SDM and QDM handle days with zero rainfall very differently, and 3) SDM more accurately accounts for the differences in the modelled variances, for temperature, between the period of interest and the calibration period.

**3.1 Stationarity and quantile mapping**

Independent of downscaling, bias correction with QM is well known to alter the raw modelled climate change signal (Hagemann et al., 2011; Themeßl et al., 2012; Brekke et al., 2013; Maurer et al., 2013; Pierce et al., 2013; Maurer and Pierce, 2014). This alteration of the raw modelled climate change signal can be attributed to the stationarity assumption which implies that the error correction values established in a calibration period can be applied to any time period within or
outside the calibration time period. These error correction values can differ, in both magnitude and sign, as a function of quantile. In the context of a warming climate, raw model projected temperature values will result in QM disproportionately sampling error correction values from higher quantiles that were established in the calibration period. Depending on whether the error correction values at the higher quantiles are greater or lesser than those at lower quantiles, this will inflate or deflate the raw model projected climate change. For example, consider a model (in the calibration period) has an uncorrected (raw)
value of 10°C that corresponds to the quantile where the empirical cumulative distribution function (ECDF) equals 0.8, and a value of 0°C corresponds to the quantile where ECDF equals 0.2. While at these same quantiles (ECDFs of 0.8 and 0.2), observations are 12°C and 3°C, respectively. Furthermore, the model is projecting more future values at 10°C. As a result,



QM will deflate or under-represent the raw model projected change. This is due to the fact that QM is more often using a 2°C error correction value (12°C-10°C, at ECDF of 0.8) instead of the 3°C (3°C-0°C, at ECDF of 0.2) correction.

Altering of the raw model's climate change signal could be justified with QM if one finds that the stationarity assumption is justified. We tested the stationarity assumption by investigating whether the error correction values are independent of the calibration period. To do this, the same daily data in the future time period (2071-2100) was bias corrected via standard QM, separately for each month, using two different calibration periods (1951-1980 and 1976-2005). Figure 2 shows how sensitive QM is to using two different calibration periods to bias correct these same future values.

Figures 2a and 2b show the sensitivity for temperature and precipitation, respectively. There are instances where this sensitivity to the calibration period is nearly as large as the raw model projected mean changes. This illustrates how unstable the error correction values in QM can be. If stationarity was a valid assumption, all the map colours in Figure 2 would be much closer to grey. The calibration period largely influences the error correction values, and as a result, the stationarity assumption is invalid. As an additional test, we performed the bias correction using a parametric implementation of QM and

found the error correction values to be equally sensitive to the chosen calibration period.

        The results shown in Figure 2 have broad implication for QM. First, it shows that one cannot confidently correct a specific modelled value with a specific error correction value. As an example, a raw modelled value of 50 mm can be corrected by -15 mm using one calibration period and +5 mm in another, leading to two very different bias corrected values of 35 mm and 55 mm, respectively. The effect that the calibration period has on QM can be mitigated to some extent by

calibrating on longer historical records (Teng et al., 2015). However, one can never be sure that these error correction values have converged to be completely independent of time. Second, due to the fact that the error correction values in QM are not stationary, the altering of the climate change signal that results from this assumption is unjustified. Therefore, until some bias correction method provides proper justification to manipulate and alter the raw model projected climate change signal, a better performing method should strive to preserve the original projected changes.

**3.2 Parametric versus non-parametric methodological approaches**

In a non-parametric method such as QM, there is an implicit assumption that each respective quantile is equally probable. In other words, the largest observed and modelled quantiles both correspond to the same empirical cumulative distribution function (ECDF) value. Therefore, is it safe to assume that events that share the same ECDF value are equally probable? To test this assumption, synthetic precipitation data was used to evaluate if events of equal quantiles can be treated as being

equally probable. Figure 3 shows two distributions (referred to as Obs and Mod), each consisting of 200 values randomly sampled from the same gamma distribution with shape parameter equal to 0.8 and scale parameter equal to 12.0. Figure 3a shows the empirical distributions, while Figure 3b shows the fitted distributions of the same data. In this example, the largest quantiles from Obs and Mod are 32.8 and 57.3. In a non-parametric method like standard QM, the correction applied to the largest Mod quantile (to bias correct this quantile) is simply the difference between the largest quantile from Mod and Obs





(depicted by the blue line with arrows). The absolute value of this difference is 24.5. When gamma distributions are fit to the empirical Obs and Mod data, it is found that the largest quantile of Obs (with the value of 32.8) corresponds to a fitted CDF of 0.993 while the largest quantile of Mod (with the value of 57.3) corresponds to a fitted CDF of 0.998. Using the fitted distributions, one can find the corresponding Mod value at the same expected Obs CDF of 0.993. In this example, that value is 44.8. Then our expected difference between events of the same expected probability has been reduced from 24.5 to 12.0

(where 24.5 = 57.3-32.8 and 12.0 = 44.8-32.8), depicted again as the blue line with arrows in Figure 2b. With this one example case, it is impossible to know if we are truly gaining information by accounting for differences in event likelihood.

Figure 4 shows the results of a 1000 randomly generated Obs and Mods values for distribution sizes of 100 and 10,000. Again, the values are randomly sampled from the same gamma distribution with shape parameter equal to 0.8 and scale parameter equal to 12.0. Figures 4a and 4d show the possible scenarios of ECDFs for the different distribution sizes.

Figures 4b and 4e show the counts of the absolute differences between the extreme values using the non-parametric method (blue line) and the parametric method (green line). Similarly, Figures 4c and 4f show the counts of the absolute differences, averaged over the entire distribution, between the values using the non-parametric and the parametric method. One can clearly see the usefulness of a parametric method over a non-parametric method as the sample size increases. With a sample size of 10,000 values, there can still be large differences between the most extreme quantiles of each distribution. However,

with larger sample sizes, we are converging on identical distributions for Obs and Mod. Therefore, the magnitude of the differences between extreme quantiles, is simply due to sampling noise. On the other hand, if a distribution is fit to the data first, then one gains information regarding the expected probabilities of specific events taking place with respect to the underlying distribution. This allows the parametric method to reduce the error associated with sampling noise over that of the non-parametric method.

## 3.3 Validating bias correction methods

Split-sample or cross-validation tests are commonly used to validate how well bias correction methods perform under changing conditions (Maurer and Pierce, 2014; Klemeš, 1986; Wang and Chen, 2014; Piani et al., 2010). Typically, a period is chosen for calibrating the bias correction parameters and then different bias correction methods' performances are compared in a validation period. For example, bias correction parameters might be fit in a calibration period such as 1951-

1980. Then, the performances of different methods are evaluated by comparing the bias corrected data to observations for the period 1981-2010. Unfortunately, split-sample tests that directly compare observations to bias corrected model data, for a time period outside of calibration, cannot distinguish between bias correction methodological performance and the performance of the underlying raw model. In this study. we define model performance by how well the raw model simulates changes to the statistical distribution of a climate variable with respect to observed changes to the distribution.

The following example is used to illustrate how split-sample tests are not suitable to validate bias correction methods. Standard QM was used to bias correct daily values of June temperature for the period 1981-2010 (validation



period) after calibrating on the period 1951-1980. In this example, evaluation of performance was measured by the mean absolute error (MAE) between observed and bias corrected quantiles. Lower values of MAE indicate better performance and reflect better distributional agreement between the bias corrected and observed values, averaged across all quantiles. Figure

5a shows a Q-Q plot corresponding to the grid cell marked by the black X in Figures 5b and 5c. These are the temperature quantiles, in the validation period, of the bias corrected modelled data and the observed data for that specific grid cell. One can observe that the bias corrected modelled data is overestimating the temperature in this grid cell with respect to observations. The MAE for this grid cell is 1.5°C (average absolute difference between all quantiles). Figure 5b shows the MAE values across all grid cells in the European domain. Figure 5c shows the model performance error of the mean, which

is the difference between the raw model projected mean temperature changes (1981-2010 with respect to 1951-1980) and the observed mean temperature changes (1981-2010 with respect to 1951-1980). It should be noted that the values in Figure 5c are completely independent of any bias correction method being implemented; the values are solely dependent on observed and raw model values. Figure 5d shows a scatter plot of the values corresponding to the grid cells in Figures 5b and 5c. There is clearly a strong relationship between the apparent performance of the QM, as it varies spatially, and how similar the

raw modelled projected mean change is to the observed change. Most of the variability pertaining to methodological performance (in a split-sample test) can be explained by the model performance error of the mean. If the raw modelled projected mean change in some grid cell is close to the observed change that took place (whether due to long-term forcing or internal climate variability), then it will appear that QM performs better in this grid cell, and vice-versa.

Next, the differences between the raw modelled changes and the observed changes (depicted by Figure 5c) are

removed from each grid cell. The copper scatter in Figure 5e shows the QM bias corrected data after removing the temperature model performance error of the mean (1.4°C, fuchsia X on the x-axis of Figure 5d). The model performance error of the mean was similarly removed for all grid cells. After removing the mean temperature model performance error, the MAE of the QM bias corrected data can be seen in Figure 5f. The apparent performance of the QM method can now be attributed to the model performance error of the standard deviation (Figure 5g). Again, much of the perceived method

performance is simply due to how well the raw model simulated changes to the standard deviation (Figure 5h). Figures 5i-l further adjusts the QM bias corrected data by removing both the model performance errors from the mean and standard deviation. Still, there is statistically significant relationship (p<.01) between method performance and the model performance error of the skewness.

Figure 5 illustrates how a split-sample or cross-validation test does not distinguish between methodological

performance and raw model performance. Using pseudo-realities (Maraun et al., 2010) could tell us something more about the robustness of methods in different scenarios, but the performance of the bias correction method still cannot be separated from how well individual models simulate raw projected changes relative to the other models. In our example, we only looked at QM method performance and how this relates to model performance. Obviously, there will be differences from one bias correction method to another. However, in a split-sample test, method and model performance are conflated. Consider a

case, where a raw model projects changes across Europe that are 1°C less than what was observed. Additionally, a particular



bias correction method inflates these projected changes by 1°C, thereby cancelling out the model performance error of the mean. It would appear that this particular method is performing better simply due to influence of model performance on validation.

Part of the validation procedure must ensure that the observed and modelled distributions in the calibration or
historical period are statistically similar (Maraun et al., 2015). However, as shown here, validating bias corrected data in another period outside of the calibration period against observations will not isolate bias correction methodological performance. Model performance will obscure the performance of the bias correction method. Instead, validation (or evaluation) should measure how well the raw modelled projected changes to the entire distribution are captured or preserved by the bias correction method between any two periods.

## 4 Scaled distribution mapping: Method description and performance

Previous sections have shown that QM lacks justification for introducing inflation/deflation to the climate change signal (Sect. 3.1). This section introduces a bias correction method named scaled distribution mapping (SDM) and its performance is compared to standard QM in addition to more resent and similar methods such as DETQM and QDM.  Methodological performance is evaluated using raw model projected changes to the leading moments (mean, standard devaition, and
skewness) instead of individual quantiles because of our findings concerning the sampling noise associated with extreme values in the distributions (Sect. 3.2).

### 4.1 Scaled distribution mapping

A new bias correction methodology, called scaled distribution mapping (SDM), is proposed in this study. The conceptual framework of the method is quite similar to QDM (Figure 1). However, as previously mentioned, our method has important
differences that will be discussed in more detail in Performance (Sect. 4.2). The SDM method makes no assumption of stationarity. It scales the observed distribution by raw model projected changes in magnitude, rain-day frequency (for precipitation) and likelihood of events. The scaling changes as a function of the bias correction period. The next two subsections outline the SDM methodology for precipitation and temperature. Similar to other bias correction methods, a pre-screening of appropriate GCMs/RCMs is advised. Bias correction will not, and should not be expected to, fix serious model
deficiencies (garbage in - garbage out, e.g., Noguer et al., 1998). There are a few important differences between the implementation of SDM for precipitation and temperature. First, SDM scales the distribution of precipitation by a multiplicative or relative amount and temperature is scaled by an absolute amount. Second, only values of positive precipitation exceeding a specified threshold (e.g., .1 mm) are used to build the distributions, while with temperature all values are used. Third, temperature data is first detrended, then bias corrected, and finally, the trends are added back in. As a
result, the variance is not inflated by temporal trends.




### 4.1.1 Precipitation

The SDM methodology for bias correcting daily precipitation is illustrated by the example in Figure 6. The observed and modelled data are from the same grid cell in the European domain for the month of April. The observed and historical model periods correspond to 1971-2000. The future model period is 2071-2100. The future period can more generally be thought of as any time period one desires to bias correct, and it can be the same period as the raw historical model period. The SDM methodology for precipitation is outline in the following steps.

Step 1) Set a raw modelled minimum precipitation threshold. In this study, we have used 0.1 mm as a threshold (which is the minimum amount of observed precipitation). Any values below the threshold are set to 0.0 mm. Next, the days with precipitation are separated from days with no rainfall. Figure 6a shows the sorted values of precipitation for the observations, the raw historical model, and the raw future model. In this example, there are 434 observed rain days, 525 raw historical model rain days, and 593 raw future model rain days. Our expected number of bias corrected future model rain days, $RD_{BC}$, can be defined as:

$$RD_{BC} = RD_{MODF} \times \frac{RD_{OBS}/TD_{OBS}}{RD_{MODH}/TD_{MODH}} ,$$

(1)

where $TD_{OBS}$ and $TD_{MODH}$ are the total number of days including non-rain days for observations and raw historical model, while $RD_{MODF}$, $RD_{OBS}$ and $RD_{MODH}$ are the number of rain days for the raw model future, observations and raw historical model. In this example, the total number of days including non-rain days is 900 for both the observations and the raw historical model (30 years times 30 days in April). These lengths of days are included to allow the flexibility of different calibration period lengths. Then, $RD_{BC}$ is found to be 593*(434/900)/(525/900) = 490 days (490 is the nearest integer value).

Step 2) Fit gamma distribution parameters, using maximum likelihood, to the positive observed precipitation values (Figure 6b), and the raw historical and future modelled values (Figure 6c). The probability density function of the gamma distribution is:

$$f(x;k,\theta) = \frac{x^{k-1}\exp(-x/\theta)}{\theta^k \Gamma(k)} ,$$

(2)

where k(>0) is the shape parameter, θ(>0) is the scale parameter, x(>0) is the precipitation amount, and Γ(k) is the gamma function evaluated at k. Next, use the fitted shape and scale parameters to find the corresponding CDF values of the positive precipitation events in the three time series. Set an upper threshold for the CDF values (e.g., 0.9999999) since imprecise rounding can lead to the cdf function providing a value of 1.0, which corresponds to an infinite precipitation amount.



Step 3) Calculate the scaling between the fitted raw future model distribution and the fitted raw historical distribution at all of the CDF values corresponding to the precipitation events of the raw future model time series. The scaling is calculated as:

$$SF_R = \frac{ICDF_{MODF}(CDF_{MODF})}{ICDF_{MODH}(CDF_{MODF})} \quad ,$$

(3)

where $SF_R$ is an array of relative scaling factors (the length is equal to number of rain days in the raw future model, which in this case is 593 values), $ICDF_{MODF}$ and $ICDF_{MODH}$ are the inverse cumulative distribution functions (ICDFs), or the percent point functions, for the fitted future and historical model distributions, respectively, while $CDF_{MODF}$ are the estimated CDF values for the future raw model corresponding to the fitted distribution. The relative scaling factors, for each raw future modelled value, can be seen in Figure 6d. As an example, lets find the scaling factor that

would correspond to the largest value in the raw future model time series. In the raw future model time series, this value is 35.8 mm. Using the fitted raw future model distribution, this event corresponds to a CDF value of 0.9974. The value $ICDF_{MODF}(.9974)$ will yield the original value of 35.8 mm, while $ICDF_{MODH}(.9974)$ is equal to 30.6 mm. The most extreme value that is bias corrected will have a relative scaling factor equal to 1.17 (35.8 mm / 30.6 mm). For reference, the largest value in the raw historical model time series is 40.6 mm (seen in Figure 6a). That value is more

extreme with respect to its own distribution, with a corresponding CDF value of .9995. However, we want to compare events that are equally probable (as discussed in Section 3.3).

    Step 4) Calculate the return periods for the three sorted arrays of positive precipitation. The return period array, RP, is calculated as:

$$RP = \frac{1}{1 - CDF} \quad ,$$

270    (4)

where CDF is the array of values found in Step 2. Proceeding with the values from the previous step, the largest modelled event in the future period had a corresponding CDF value of 0.9974, which corresponds to a return period of 385 days (seen as the largest value of the blue line in Figure 6e). Similarly, the largest return periods of the observations and raw historical model are 1667 and 2000 days, respectively. To compare the return periods across the entire distribution, the observed and

raw historical model RPs are linearly interpolated. Figure 6e shows the linearly interpolated RPs (the linear interpolation stretches or contracts the values along the x-axis, keeping the original range).

    Step 5) Find the scaled or adjusted RP for the raw future model. This is calculated as:

$$RP_{SCALED} = maximum\left(1, \frac{RP_{IOBS} \times RP_{MODF}}{RP_{IMODH}}\right) \quad ,$$



(5)

where $RP_{MODF}$ is the RP for the raw future model and $RP_{IOBS}$ and $RP_{IMODH}$ are the linearly interpolated RPs for the observations and raw historical model, respectively. The maximum value, which is greater than or equal to 1, is used to evaluate each value in the $RP_{SCALED}$ array. This is necessary (especially for temperature) to ensure that the values of $CDF_{SCALED}$ in Eq. (6) are between 0 and 1. $RP_{SCALED}$ is shown in Figure 6f as the gold line. This modifies or scales the RP of observed events by the projected changes to the extremity of modelled events. $RP_{SCALED}$, for the most

extreme value, is found to be 321 days = 1667*385/2000. As a result, the return period of the most extreme observed value is reduced because the raw future modelled extreme value was more likely (smaller return period) than that of the raw historical modelled extreme value. Use the $RP_{SCALED}$ to find the corresponding scaled CDF values with:

$$CDF_{SCALED} = sort\left(1 - \frac{1}{RP_{SCALED}}\right) \quad,$$

(6)

where $CDF_{SCALED}$ is the scaling of the modelled change in event likelihood with respect to the observed likelihoods. The array is sorted in ascending order.

Step 6) The initial array of bias corrected values can now be calculated as:

$$BC_{INITIAL} = ICDF_{OBS}\left(CDF_{SCALED}\right) \times SF_R \quad,$$

(7)

where $ICDF_{OBS}$ is the inverse cumulative distribution function for the observed fitted distribution and $CDF_{SCALED}$ and $SF_R$ are obtained from Eqs. (6) and (3). Figure 6g shows $BC_{INITIAL}$ as the gold line. Lastly, the frequency of rain days needs to be adjusted. Recall from Eq. (1), $RD_{BC}$ is equal to 490 days. $BC_{INITIAL}$ is linearly interpolated from a length of 593 days to 490 days. This yields the bias corrected values which can be seen as the black line in Figure 6h.

Step 7) As a final step, the bias corrected values for positive rain days, are placed back into the modelled time

series in the correct temporal locations. Consider in this example that the maximum raw modelled precipitation amount fell on April 28, 2078. Then, the largest bias corrected value will be reinserted back into that day. This is applied to the rest of the 489 positive values of precipitation. Originally, there were 593 raw modelled future rain days, and therefore, the smallest 103 raw modelled values will have no precipitation after bias correction. Similar to this example, GCMs/RCMs more often overestimate the frequency of rain days (Leander et al., 2008). However, in the case of the model underestimating the rain

day frequency, the SDM method is currently not adjusting the original raw modelled rain day frequency. It should be kept in mind, however, that even if the model underestimated the frequency, the impact on the distribution is significantly less. In this example, if the raw model had 10% fewer rain days than observations, that would have only translated to a 0.4% reduction in total precipitation. This is due to the fact that the method is preferentially filling the largest events first.


In the case that the historical model period and the bias correction period of interest completely overlap, the bias
corrected data will be exactly the same as the observed distribution. There would be no difference between the distributions
in magnitude, likelihood, nor rain-day frequency, and therefore the observed distribution undergoes no scaling.

### 4.1.2 Temperature

The SDM methodology for temperature is outline in the following steps. Again, the future period is a general representation
of any time period one desires to bias correct, and it can be the same period as the raw historical model period.

Step 1) Detrend the raw modelled and observed time series in order to get a more accurate measure of the natural
variability (we have used a linear trend, though any trend line could be used). These trends are added back in at the end, but
until then, all subsequent steps use the detrended time series.

Step 2) Fit a normal probability distribution function to the detrended observed, raw historical modelled, and the
raw future modelled time series. For a normal distribution, the fitted parameters are simply the empirical mean and standard
deviation. Next, using these fitted normal distributions, find the corresponding CDF values for the temperature events that
occurred in the three time series. Similarly to precpitation, set an upper and lower threshold for the CDF values (e.g., 0.0001,
0.9999). Again, imprecise rounding can lead to the cdf function providing a value of 1.0 or 0.0, which corresponds to an
infinite temperature values.

Step 3) Calculate the scaling between the fitted raw future model distribution and the fitted raw historical
distribution at each probability of the events taking place in the raw future model time series. The scaling is then calculated
as:

$$SF_A = \left[ ICDF_{MODF}\left(CDF_{MODF}\right) - ICDF_{MODH}\left(CDF_{MODF}\right) \right] \times \left( \frac{\sigma OBS}{\sigma MODH} \right) \, ,$$

(8)

where $SF_A$ is an array of absolute scaling factors, $ICDF_{MODF}$ and $ICDF_{MODH}$ are again the inverse
cumulative distribution functions (ICDFs) for the fitted future and historical model distributions, $CDF_{MODF}$ is an array
with the estimated CDF values for the future raw model corresponding to the fitted distribution, and $\sigma OBS$ and
$\sigma MODH$ are the standard deviations of the observed and raw historical distributions.

Step 4) Next, calculate the return periods for the three sorted arrays of temperature with:

$$RP = \frac{1}{\left[ 0.5 - \left| \left( CDF - 0.5 \right) \right| \right]} \, ,$$

335    (9)

Eq. (9) is different from Eq. (4) to reflect the two-tailed nature of a normal distribution.



Step 5) Find $RP_{SCALED}$ for the raw future model using Eq. (5). If the lengths of the time series are the same for the historical and future periods, no linear interpolation is required. Then, use the $RP_{SCALED}$ array to find the corresponding modified CDF values with:

$$CDF_{SCALED} = sort\left[ 0.5 + sign\left(CDF_{OBS} - 0.5\right) \times \left\| \left(0.5 - \frac{1}{RP_{SCALED}}\right) \right\| \right] \,,$$

(10)

where $sign\left(CDF_{OBS} - 0.5\right)$ is an array corresponding to the sign of the values (this will be an array of ones and negative ones).

Step 6) The initial array of bias corrected values can now be calculated as:

$$BC_{INITIAL} = ICDF_{OBS}\left(CDF_{SCALED}\right) + SF_A \,,$$

(11)

where all variables have been previously defined.

Step 7) Reinsert the bias corrected values, $BC_{INITIAL}$, back into the correct temporal locations from the original raw future modelled time series. Lastly, the trend of the raw future modelled time series is added back into the bias corrected time series. Like with precipitation, when the historical model period and the bias correction period of interest completely overlap, the bias corrected temperature data will be exactly the same as the observed distribution (except the trend of the bias corrected data will be that of the modelled trend and not of observations).

### 4.1.3 SDM and the temporal evolution of climate change

The SDM method presented in subsections 3.6.1 and 3.6.2 attempts to best preserve the raw model projected changes to different moments of the distribution. However, the temporal evolution of the climate change signal might not be captured. Consider the case where SDM is used to bias correct January temperature values for the period 2011-2100 using the calibration period 1971-2000. Then, the changes to the bias corrected distribution of 2011-2100 versus 1971-2000 will be very similar to the raw model projected changes to the distribution. On the other hand, investigating a temporal subset of the bias correction period can yield undesirable results. For example, the mean change between 2011-2040 and 1971-2000 might not be as close for the bias corrected and the raw data. If one desires to have the climate change signal properly preserved across a variety of time scales, the SDM method must be discretized into smaller blocks. For this study, the authors used 30-year periods with a 10-year sliding window in order to bias correct the middle 10-years. We began by bias correcting the period 2011-2020 using the period 2001-2030 as our period of interest (future period) and 1971-2000 as our calibration period. Next, we bias corrected 2021-2030 using the period 2011-2040 (period of interest) along with the same calibration period. With no modelled data beyond 2100, the period 2091-2100 used 2081-2100 as the period of interest. A user can




adjust the amount of years to bias correct length of the period, amount of years to save and the length of the sliding window to more/less strongly follow the raw modelled temporal evolution of climate change.

## 4.2 Performance

Figure 7 illustrates the amount of inflation (or deflation) that was introduced to the raw model projected change to the mean
by the SDM and standard QM methods. The mean climate change was evaluated between the periods 2071-2100 and 1971-2000. Figures 7a and 7b show the difference, by month, between the raw mean temperature changes and the bias corrected mean temperature changes when using SDM and QM, respectively. Similarly, Figures 7c and 7d show the same but for relative change to precipitation. For both temperature and precipitation, SDM has minimal inflation to the raw model projected mean change. In contrast, using QM leads to inflation greater than 1°C and 10%, for temperature and precipitation
respectively, across large regions of Europe. Figure 8 is the same as Figure 7, but depicting the inflation of the standard deviation for SDM versus QM. Again, SDM much better preserves the raw model projected changes to the standard deviation.

The temporal evolution of the SDM and QM performance, for all grid cells, is illustrated for temperature in Figure 9. As discussed in subsection 4.1.3, the authors implemented SDM using 30-year periods with a 10-year sliding window in
order to bias correct the middle 10-years. The amount of inflation/deflation to the climate change signals is illustrated for the mean, standard deviation, skewness, and the trend. Each coloured grid cell in the figure depicts the spatial mean absolute error (MAE) between the raw model and bias corrected changes for all grid cells. For example, consider the lower left grid cell situated in the SDM column and Mean row. This grid cell shows the spatial MAE between the raw modelled and bias corrected mean changes between the periods 2071-2100 and 1971-2000 for the month of January (it is the spatial MAE of
the January subplot of Figure 7a). What is most noticeable is how the QM's alteration (inflation/deflation) of the climate change signal increases as a function of the projected time period. When the projected period is furthest from the calibration period (2071-2100), the alteration to the leading three moments are the greatest. In contrast, SDM is seen to outperform QM in its ability to better preserve the raw projected changes. Furthermore, the performance of SDM does not degrade as a function of the projected time period. Figure 10 shows the same as Figure 9, but for precipitation. Again, SDM performs
much better in preserving the raw projected changes. This is especially true for projected changes to the mean and the standard deviation. It should be noted that all methods perform worse for precipitation in the summer months. This can be explained by some regions (e.g., Spain) having very few days with precipitation, which cannot be fit well by either a parametric or a non-parametric method.

Figure 11 compares the temperature performances of SDM to the more recent and similar methods of QDM and
DETQM. SDM is seen to perform best with respect to better preserving the raw projected changes for the leading three moments of the distribution. Both SDM and QDM perform equally well for preserving changes to the mean. However, SDM better minimizes MAE for standard deviation and skewness. This can be attributed to one of the main differences between





the SDM and QDM methods. QDM scales the observed distribution by absolute difference between modelled quantiles (future - past), though, this does not properly scale the higher moments of the distribution. In contrast, SDM applies the

rightmost term, (σOBS / σMODH), in Eq. (8). This results in more appropriately scaling the higher moments of the bias corrected temperature data. The other difference between the methods is that SDM is parametric, while QDM is non-parametric or empirical.

    Figure 12 compares the performances of the same three methods, but for precipitation. Again, SDM outperforms QDM and DETQM. In the case of precipitation, though, SDM shows the greatest improvement in its ability to preserve the

raw projected mean change. This can be explained by the different ways that SDM, QDM and DETQM handle days with zero precipitation. DETQM removes the mean modelled trend, but still performs poorly because the detrended error correction values are still assumed to be stationary. Like SDM, QDM implements a threshold of modelled precipitation. Then, these days of zero precipitation are filled with non-zero uniform random values below the trace threshold prior to bias correction. The multiplicative scaling amounts are subsequently found between all quantiles and applied to the observed

quantiles. After bias correction, the values below the trace threshold are set back to zero. With that approach, the scaling is unstable when there is a mismatch in rain-day frequency. For example, consider a simplified example where the observed, raw future model and raw historical model have sorted arrays of precipitation amounts in mm of [0,1,4,15], [0,1,3,10], [0,0,1,8], respectively. After filling with uniform non-zero amounts, assume these arrays become [.02,1,4,15], [.04,1,3,10], [.02,.04,1,8]. The scaling array would then be [2,25,3,1.25] = [.04,1,3,10]/[.02,.04,1,8]. Multiplying the scaling array by the

observed array gives [.04,25,12,18.75]. The raw mean change would then be 1.56 = mean([0,1,3,10])/mean([0,0,1,8]), while after bias correction it is 2.79 = mean([0,25,12,18.75])/mean([0,1,4,15]). This simplified example illustrates how implementing QDM with a mismatch of the rain-day frequency can alter the raw modelled mean change. As shown in the SDM methodology, linear interpolation is used to address the issue of different rain-day frequencies. This scales similar parts of the distribution and more explicitly changes the number of bias corrected rain-days, and as a result, SDM much better

preserves raw modelled changes to the mean.

## 5 Conclusions

Bias correction methods vary considerably and can have a large influence on the expected regional impacts of climate change. Multiple studies have previously come to the conclusion that QM is one of the better existing bias correction methods (Gudmundsson et al., 2012; Teutschbein and Seibert, 2012; Teutschbein and Seibert 2013; Chen et al., 2013).

However, our analysis highlighted two issues that challenge this conclusion. First, we demonstrated that the stationarity assumption is invalid and, as a result, the climate change signal cannot be justifiably altered using QM. Second, split-sample or cross-validation evaluation tests do not isolate the performance of the bias correction methods themselves. These performances are conflated with how well the raw model(s) simulate the observed changes to the leading moments of the





distribution. In light of these issues, the performance of a bias correction method should be validated on how well it
preserves raw model projected changes across the entire distribution.

In this study, we presented the SDM bias correction methodology which scales the observed distribution by raw
model projected changes to magnitude, rain-day frequency (for precipitation) and the likelihood of events. The performance
of SDM was evaluated and found to perform better than traditional QM along with recent methods that are more similar such
as QDM and DETQM. We advocate using a bias correction method, like SDM, which scales the observed distribution by
simulated changes across the modelled distribution. As a result, one need not rely on the invalid stationarity assumption.

**Data and code availability**

The data is available at ESGF (e.g., https://pcmdi.llnl.gov/projects/esgf-llnl/) and the python codes for running the SDM
methodology are available upon request from the corresponding author. A reference implementation can be obtained from
https://github.com/wegener-center/pyCAT.

**Competing interests**

The authors declare that they have no conflict of interest.

**Acknowledgments.** The work was supported by the project ÖKS15, funded by the Austrian Federal Ministry of Agriculture,
Forestry, Environment and Water Management. The authors had additional support from the United States Bureau of
Reclamation, and the federal state government of Austria with the projects CHC-FloodS (id: B368584) and DEUCALION II
(id: B464795), both funded by the Austrian Klima- und Energiefonds through the Austrian Climate Research Program
(ACRP). We acknowledge the World Climate Research Programme's Working Group on Regional Climate, and the Working
Group on Coupled Modelling, former coordinating body of CORDEX and responsible panel for CMIP5. We also thank the
climate modelling groups (Royal Netherlands Meteorological Institute (KNMI)) for producing and making available their
model output. We also acknowledge the Earth System Grid Federation infrastructure an international effort led by the U.S.
Department of Energy's Program for Climate Model Diagnosis and Intercomparison, the European Network for Earth
System Modelling and other partners in the Global Organisation for Earth System Science Portals (GO-ESSP). We
acknowledge the E-OBS dataset from the EU-FP6 project ENSEMBLES (http://ensembles-eu.metoffice.com) and the data
providers in the ECA&D project (http://www.ecad.eu). The authors acknowledge the computing time granted by the John
von Neumann Institute for Computing (NIC) and provided on the supercomputer JURECA at Jülich Supercomputing Centre
(JSC). Finally, we gratefully acknowledge the helpful comments and suggestions of Douglas Maraun.





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





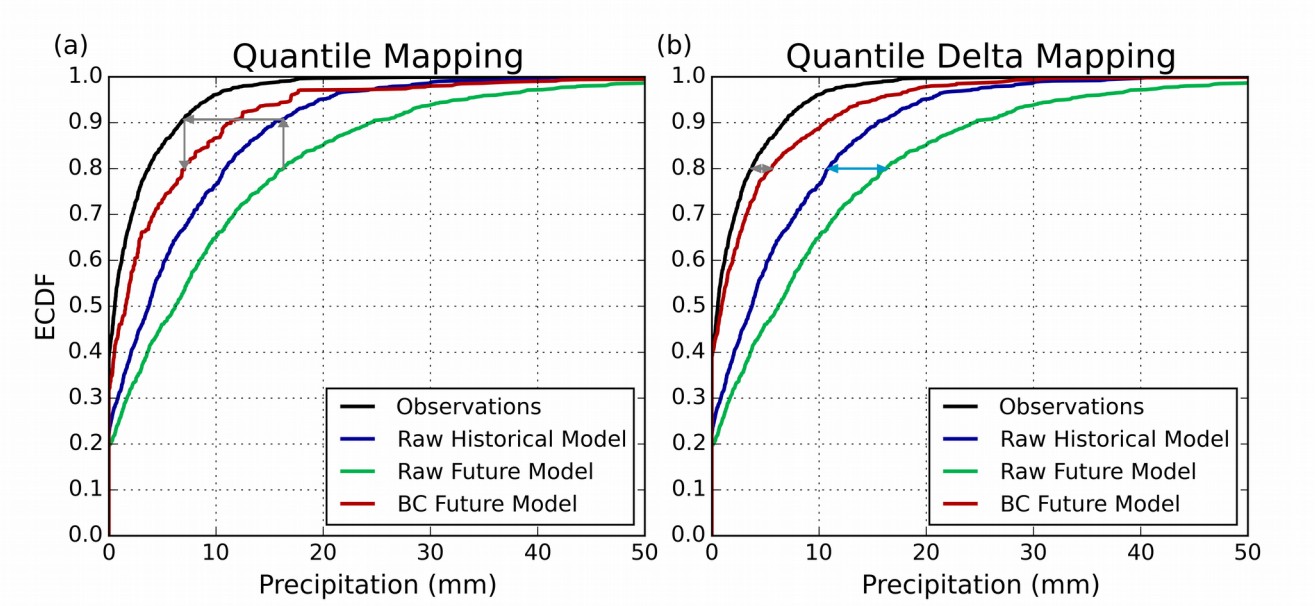

Figure 1: Schematic of the quantile mapping versus quantile delta mapping methodologies. The red lines are the bias corrected values for the future model. The arrows in each subplot illustrate the bias correction of a future modelled value at ECDF = 0.8.





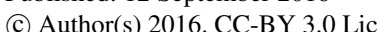

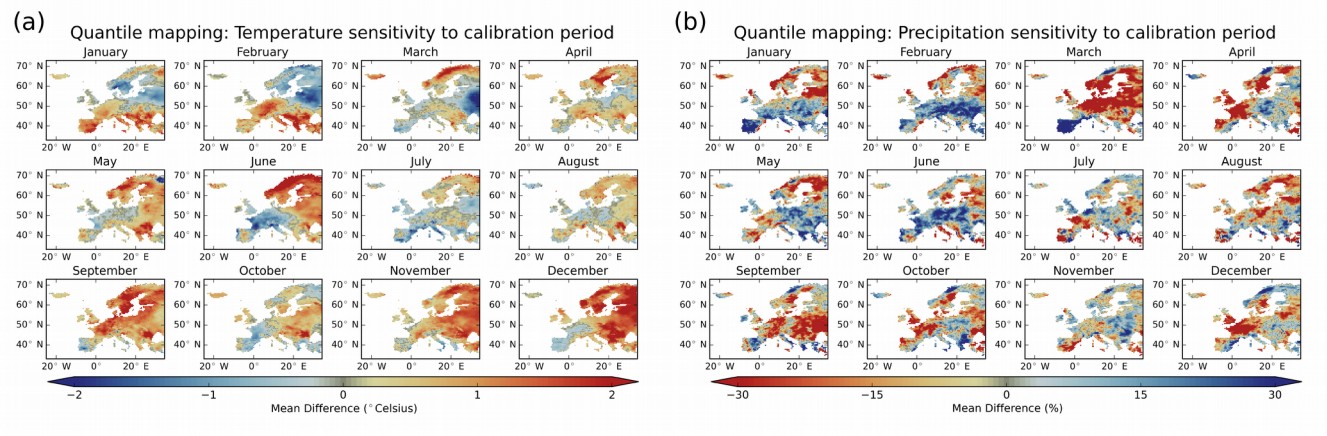

Figure 2: The sensitivity of quantile mapping (QM) to the calibration period. The same daily data in the future time period (2071-2100) was bias corrected, separately for each month, using two different calibration periods (1951-1980 and 1976-2005). Subplots (a) and (b) show the sensitivity for temperature and precipitation, respectively. Grey reflects no sensitivity of QM to calibration period, while increased saturation of warmer and cooler colors depict non-stationary error correction values used by QM.





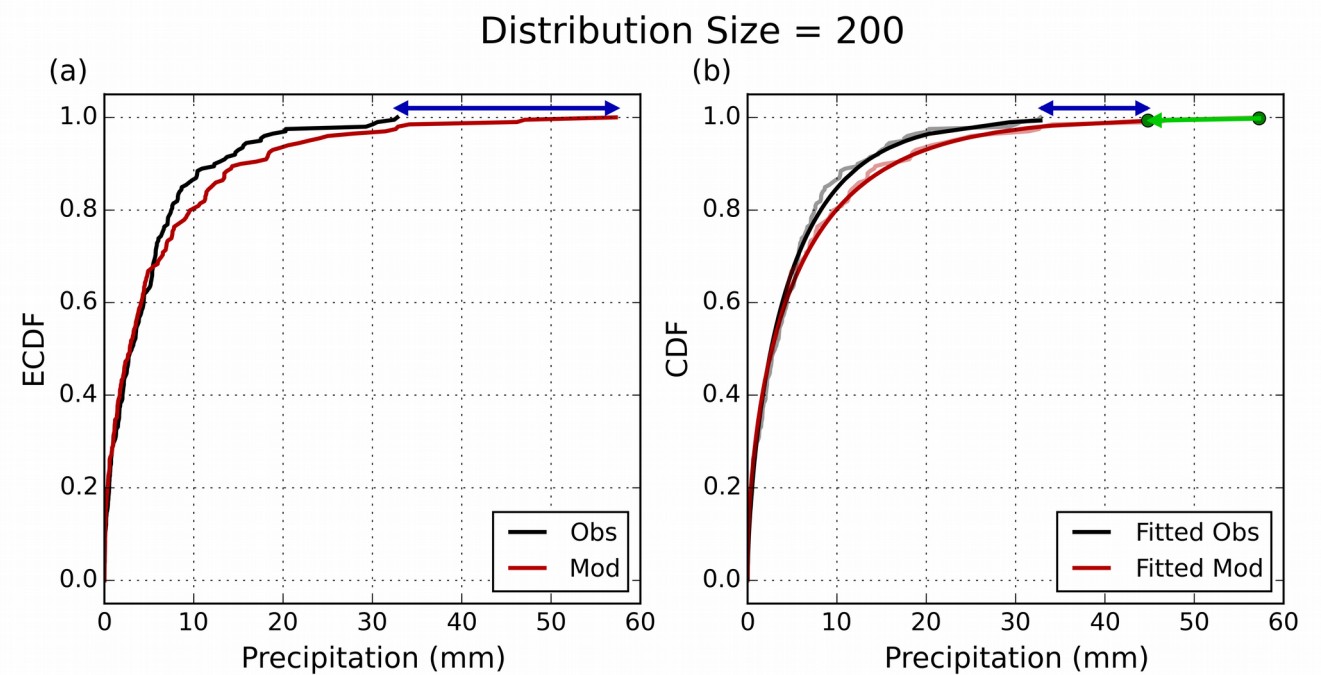

635   Figure 3: Synthetic data of observations (Obs) and modelled (Mod) data. Each data set consist of 200 values randomly sampled from the same gamma distribution with shape parameter equal to 0.8 and scale parameter equal to 12.0. Suplot (a) shows the empirical distance (blue arrow) between the largest quantile of each distribution. Subplot (b) shows the difference after fitting distributions and evaluating the largest events at the same CDF value (in this case the fitted CDF value of the observations).





Figure 4: Observed (Obs) and modelled (Mods) values for distribution sizes of 100 and 10,000. As in Figure 3, the values are randomly sampled from the same gamma distribution with shape parameter equal to 0.8 and scale parameter equal to 12.0. Subplots (a) and (d) show the possible scenarios of ECDFs for the different distribution sizes. Subplots (b) and (e) show the counts of the absolute differences between the extreme values using the non-parametric method (blue line) and the parametric method (green line). Similarly, subplots (c) and (f) show the counts of the absolute differences, averaged over the entire distribution, between the values using the non-parametric and the parametric method.




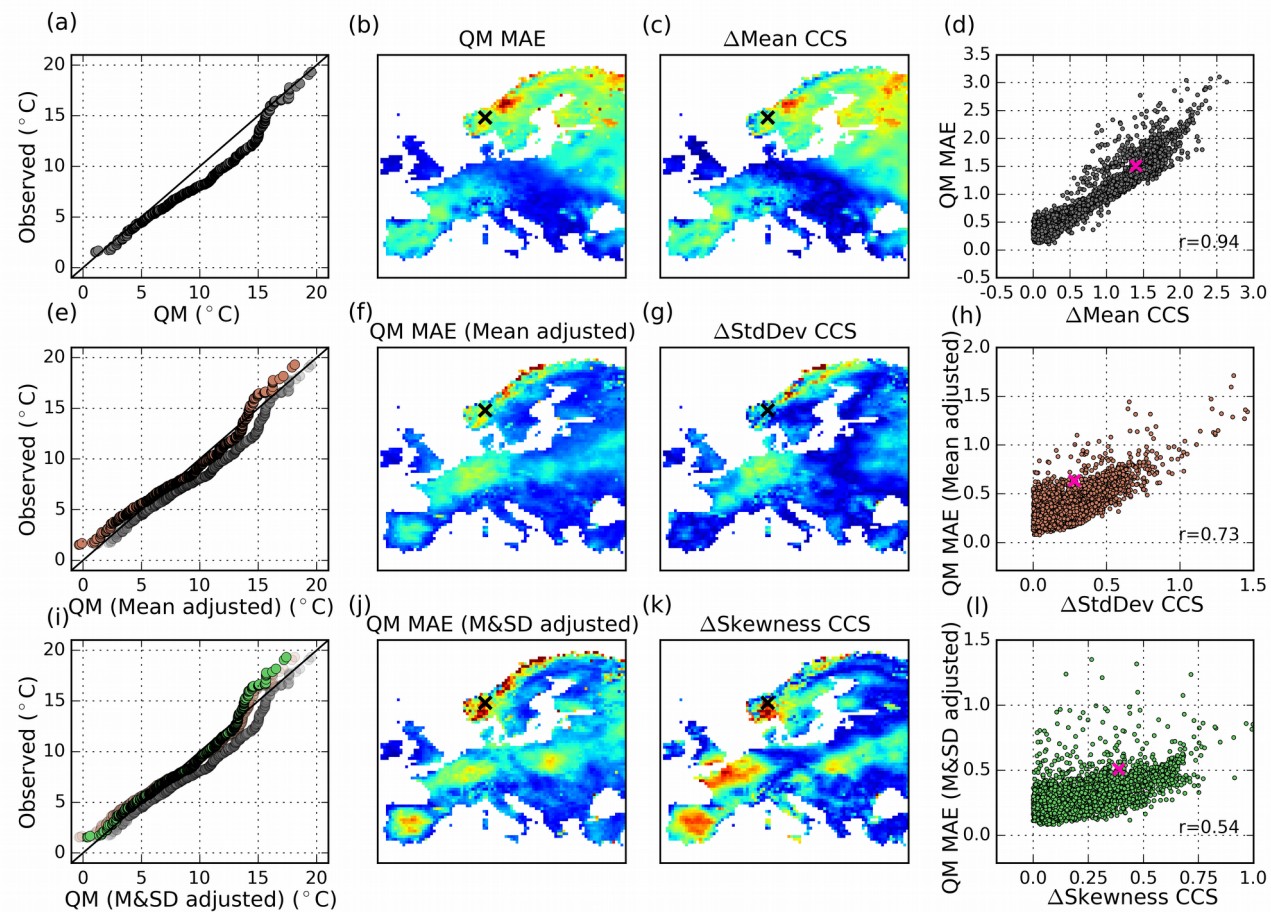

Figure 5: Conflation of apparent bias correction skill (using QM) and model performance. The data are daily temperature
values for the month of June for the period 1981-2010 after calibrating on the period 1951-1980. Suplot (a) shows a Q-Q plot
of an example grid cell delineated by the black X in the map subplots and fuschia scatter point in (d), (h) and (l). Subplot (b)
shows the mean absolute error (MAE) of the Q-Q plots from each grid cell, while (c) shows the raw model performance
error of the mean. CCS refers to the climate change signal. Suplot (d) shows the scatter and correlation between (b) and (c).
The colormaps of the second and third columns can be inferred from the scatter in the fourth column. After removing the raw
model performance error of the mean, the Q-Q plot becomes the copper scatter in (e). The QM MAE in (f) is the result of all
the raw model mean performance errors being removed from each grid cell, (g) is the raw model performance error of the
standard deviation, and (h) shows the scatter and correlation between (f) and (g). The raw model performance errors of the
mean and the standard deviation are then removed, and the Q-Q plot becomes the green scatter in (i). After removing these
errors for all grid cells, suplots (j), (k) and (l) show the relationship between QM MAE and raw model performance error of
the skewness.





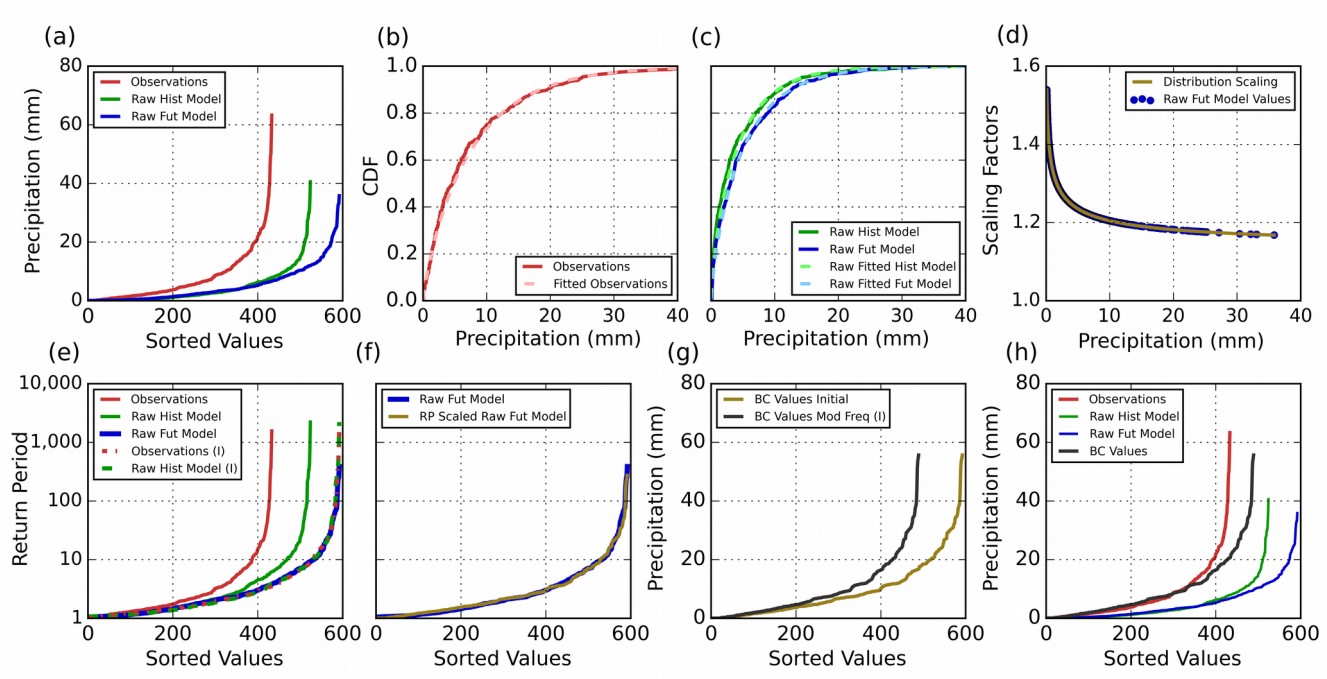

Figure 6: Illustration of the scaled distribution mapping (SDM) methodology for precipitation. Subplot (a) shows the sorted positive precipitation values for observations, along with the raw historical and raw future model. Subplots (b) and (c) show the empirical and fitted distributions. The scaling factors between the raw modelled future and historical time periods is shown in (d). The return periods and the scaled return period are plotted in (e) and (f), respectively (the (I) indicates the linear interpolation). Subplot (g) shows the initial bias corrected precipitation values in gold and the final bias corrected values for the future model period is shown as the black line in (g) and (h).




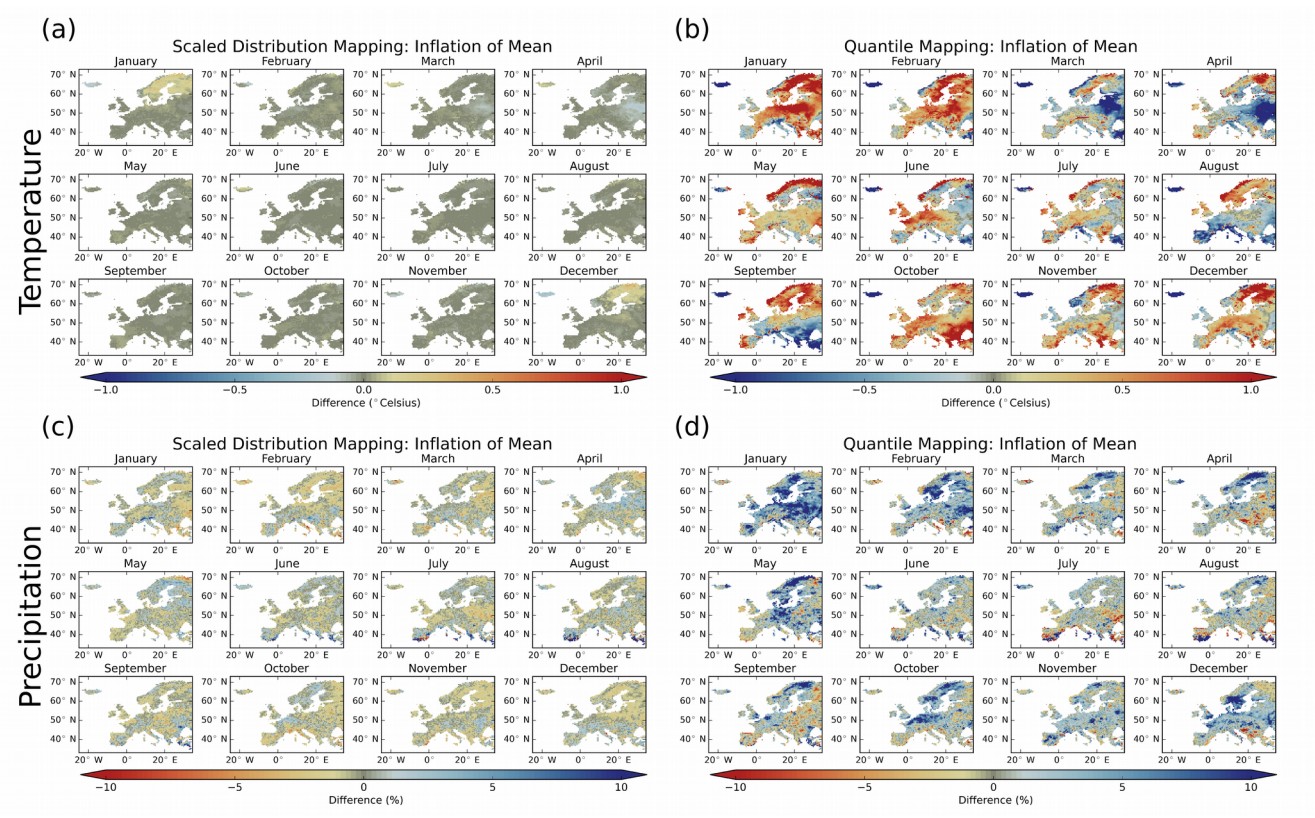

Figure 7: The amount of inflation (or deflation) introduced to the raw model projected change to the mean by the SDM and QM methods. The mean climate change was evaluated between the periods 2071-2100 and 1971-2000. Subplots (a) and (b) show the monthly differences between the raw and bias corrected mean temperature changes using SDM and QM, respectively. Similarly, (c) and (d) show the same but for relative change to precipitation.







Figure 8: Same as Figure 7, but depicting the inflation of the standard deviation for SDM versus QM.





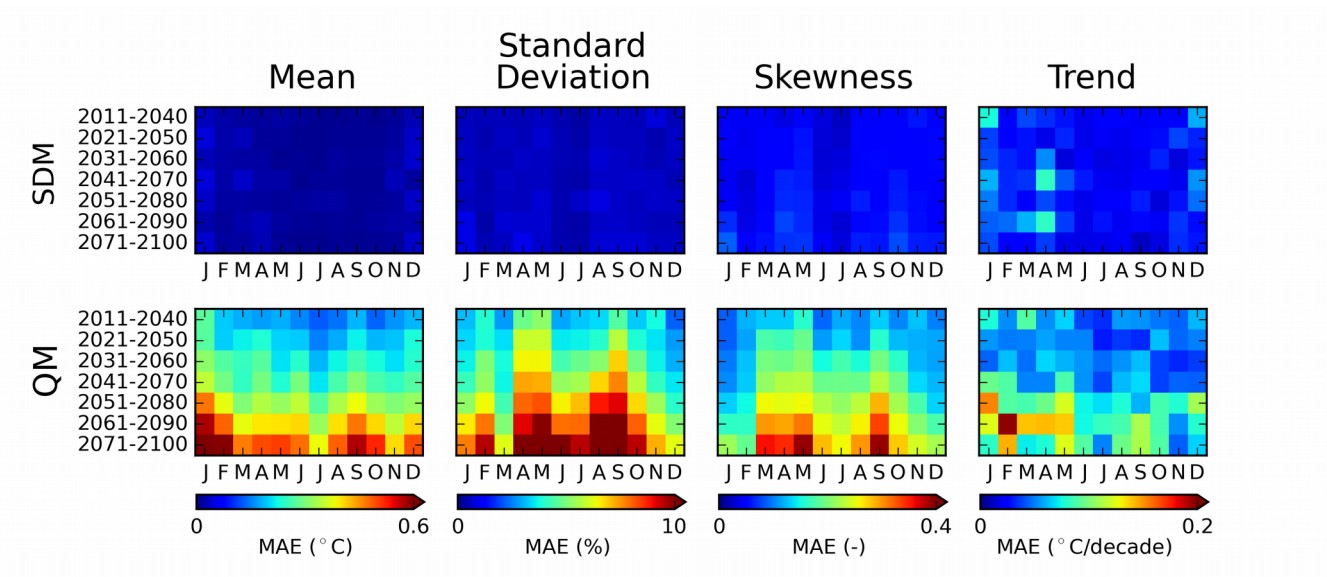

Figure 9: Performance of SDM and QM for bias correcting temperature with varying outlook periods (i.e., 2011-2040, 2021-2050, …). The colorbars correspond to the mean absolute error (MAE) between the raw model and bias corrected changes for the leading three moments of the distribution in addition to trends. The lower left grid cell situated in the SDM column and Mean row is the spatial MAE between the raw modelled and bias corrected mean changes between the periods 2071-2100 and 1971-2000 for the month of January (it is the MAE over all of Europe calculated from Figure 7a).




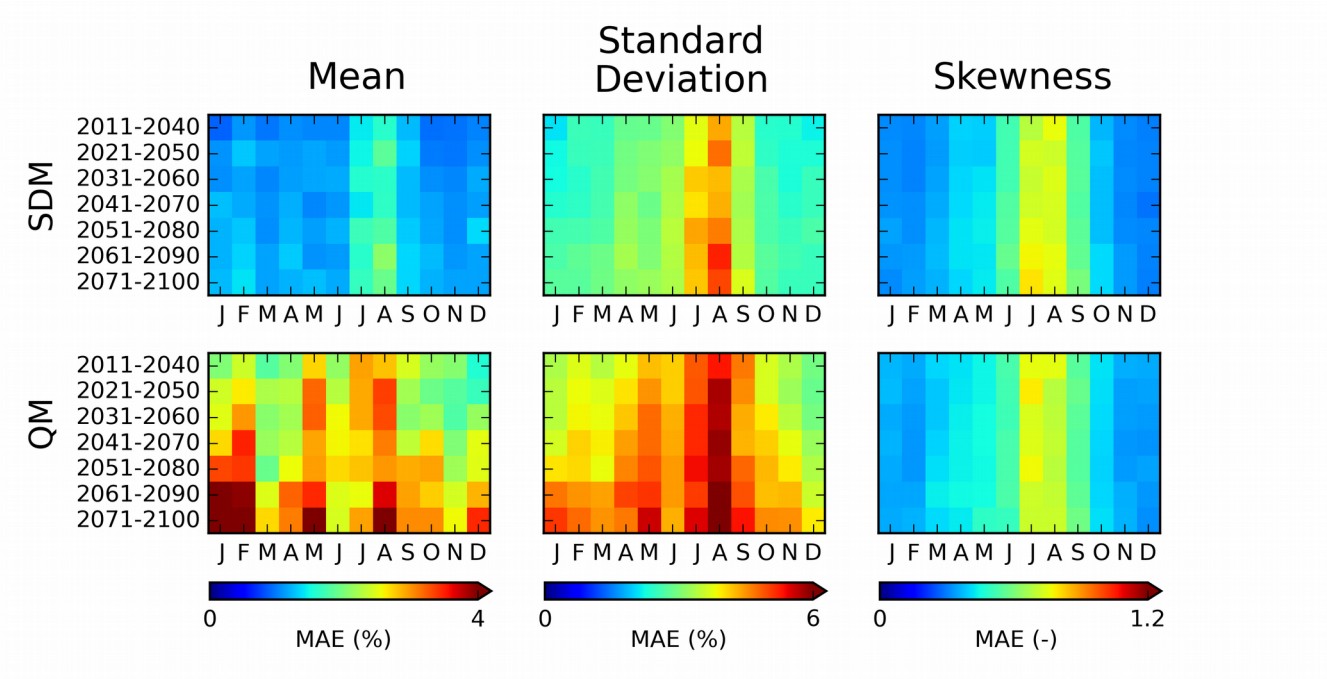

Figure 10: Same as Figure 9, but for precipitation.









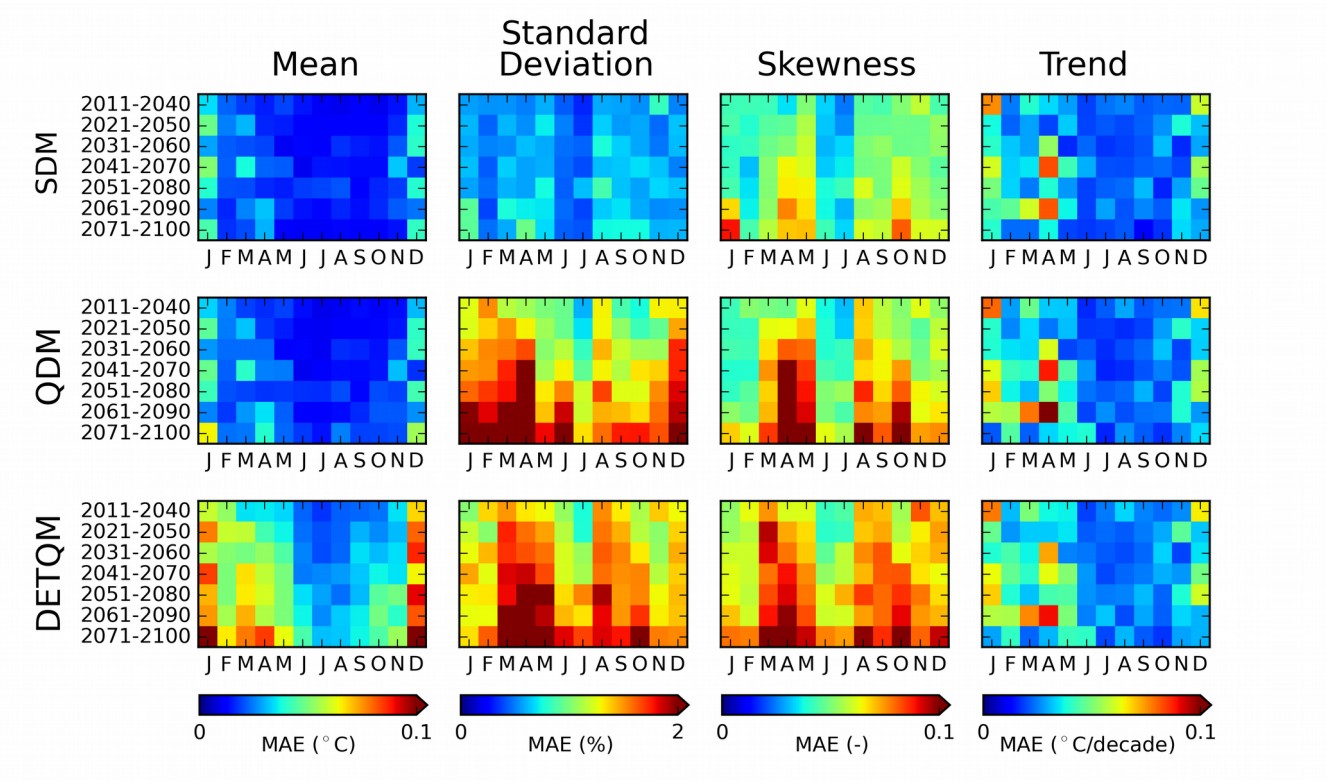

Figure 11: Same as Figure 9, but compares the temperature performances of SDM to the more recent and similar methods of QDM and DETQM.










Figure 12: Same as Figure 10, but compares the precipitation performances of SDM to the more recent and similar methods
of QDM and DETQM.
