# Peer review of "Scaled distribution mapping: a bias correction method that preserves raw climate model projected changes"

_Hydrology and Earth System Sciences, 2016_

## Referee Comment (RC1) · U. Ehret (Referee) · 10 Nov 2016

Dear Editor, dear Authors,

I have reviewed the aforementioned work. My conclusions and comments are as follows:

**1. Scope**

The article is within the scope of HESS.

**2. Summary**

The authors present three analyses related to methods of univariate bias correction methods for climate model projections, mainly related to Quantile mapping (QM) and Quantile Delta mapping (QDM) approaches, and present a new method called Scaled distribution mapping (SDM). All work presented is based on daily mean temperature and precipitation data of the KNMI-RACMO22E regional climate model and E-OBS quasi-observational data for the periods 1951-2005 and 2006-2100.

The first analysis investigates the validity of the stationarity assumption inherent to QM bias correction. The authors show, by using two different calibration periods, that the difference in correction magnitude is in the same order as the climate change signal itself. The authors conclude that stationarity hence is not a generally valid assumption for bias correction methods.

The second analysis compares nonparametric and parametric ways to compare (and bias-correct) cumulative probability density functions of meteorological variables of interest. Using cdfs sampled from gamma-distributions, the authors show that differences in value at the extreme quantiles can often be attributed to insufficient sampling size rather than real differences of the underlying distributions. The authors show that fitting theoretical distributions to the samples mitigates this effect.

The third analysis addresses the validation of bias-correction methods by split-sampling tests. Here the authors demonstrate that this approach is not suited to distinguish performance of the raw model and the bias correction method. The authors argue that evaluation should focus on whether a bias correction method preserves raw model projected changes over time.

Based on these analyses, the authors present a parametric bias-correction method related to QDM (and hence free of the stationarity assumption) called Scaled distribution mapping, which uses different approaches for temperature and precipitation data, taking into account the particularities of each variable and aims at preserving the raw model projected changes. The methods are applied in a sliding window approach to capture the temporal evolution of climate change. Comparing SDM to QM and QDM approaches reveales that the first outperformes the others with respect to preserving the raw model projected changes.

**3. Overall ranking**

The work is ranked 'Major revision'.

**4. Evaluation**

The three analyses presented by the authors have been conducted in a sound manner, the conclusions drawn by the authors are correct. The new SDM approach is a reasonable advancement of existing bias-correction methods of the QDM type.

My concerns with the study are hence not related to its direct content, but to the validity of bias-correction methods to climate change projections in general. My arguments in this respect can be found in detail in Ehret et al. (2012). In my opinion, in order to make bias correction of climate change projections a valid procedure, the following questions need to be answered:

- Reliability: How can we justify that climate models with such large deficiencies that bias correction is required are nevertheless able to correctly predict climate change?
- Physical limits: Does the bias correction avoid pushing corrected values beyond physically based limits?
- Spatiotemporal field covariance: One of the main strengths of climate models is that they provide thermodynamically consistent spatio-temporal fields of all meteorological variables. Bias correction methods applied separately to each field potentially destroys this advantage. How to prove that a given bias correction method does not do so, or how to prove that the field covariance is of minor importance for the task at hand?
- Minor role of feedbacks among variables. This is related to the above point. How to prove that the feedbacks between the variables (e.g. in land surface - atmosphere coupling), that will be potentially impaired by bias-correcting individual variables, will either not be severely altered by the bias-correction, or that the feedbacks only play a minor role?
- How can we assure that a model deficiency manifesting as spatio-temporal offset will not falsely be treated as a magnitude bias (and hence be bias-corrected)?

Please note that these points are mainly of concern in climate change studies, where the future is unknown. I have no fundamental problem with post-processing of short-term forecasts, as here stationarity of climate and model deficiencies can reasonably be assumed, which allows to link post-processing methods directly to recognized, stationary model deficiencies and to evaluate their effectiveness.

Yours sincerely,

Uwe Ehret

References

Ehret, U., E. Zehe, V. Wulfmeyer, K. Warrach-Sagi and J. Liebert (2012): HESS Opinions "Should we apply bias correction to global and regional climate model data?" Hydrol. Earth Syst. Sci. 16 (9), 3391-3404, 10.5194/hess-16-3391-2012.

---

## Author Comment (AC1) · 24 Nov 2016

Dear Dr. Uwe Ehret,

The authors would like to thank you for your valuable comments and suggestions. You have summarized well our analyses and have agreed that our proposed method is a reasonable advancement of existing bias correction methods. The authors appreciate your concerns regarding the validity of bias correction methods, and we have now included an additional paragraph highlighting these issues. The "Conclusions" section now begins:

"Bias correction methods are used extensively in impact assessment studies (Ines and

Hansen, 2006; Muerth et al., 2013; Teng et al., 2015). The application of these methods, however, is not without controversy (Ehret et al., 2012). A number of important questions that require consideration are: 1) Does independently applying bias correction to different meteorological variables (separately to precipitation and temperature) adversely alter the thermodynamically consistent spatio-temporal fields provided by climate models? 2) Do bias correction methods avoid pushing the corrected values beyond physically realistic limits? 3) Can GCMs/RCMs with large biases be reliable in their projections of climate change? 4) How can substantial model deficiencies not simply be falsely treated as bias and corrected as such? These are difficult questions, and more reflection and investigation is required before we find answers that are indisputable. In any regard, for the foreseeable future, there will continue to be scientists that use bias correction methods for impact assessment studies."

Unfortunately, we cannot address or answer all of these questions in this paper. It is outside the scope of our manuscript. Our primary contribution has been to provide a method that does not rely on faulty assumptions (found in other methods) while more accurately preserving raw climate change projections across the entire distribution. We agree that these issues you raise need to be considered. However, answering these questions is far from an easy task. We are not aware of any publications where these questions have indisputably been put to rest. It is important that the scientific community continues to explore where, when and why bias correction could or could not be valid. During this ongoing process, researchers will still make the choice to bias correct climate model output data. Our proposed method provides a more robust and justifiable approach for impact modellers to use while these more challenging questions are pursued and become more well understood.

Thank you again for your thoughtful review and for highlighting these difficult but important questions.

Yours sincerely,

Matthew Switanek and coauthors

---

## Referee Comment (RC2) · A. Bàrdossy (Referee) · 13 Dec 2016

*Review of the paper*
**Scaled distribution mapping: a bias correction method that preserves raw climate model projected changes.**
*by Matthew B. Switanek, Peter A. Troch, Christopher L. Castro, Armin Leuprecht, Hsin-I Chang, Rajarshi Mukherjee, Eleonora M.C. Demaria*
*submitted for publication in*
*Hydrol. Earth Syst. Sci.*

This paper treats an important problem - how to bias correct RCM output to be used for possible impact assessment. The authors argue that the frequently preferred quantile mapping is under circumstances leading to unreasonable results - therefore another more *robust* method is needed. The scaled distribution mapping (SDM) suggested by the authors is a sophisticated version of the classical alteration of sequences by multiplication of precipitation and linear scaling of temperature. The method is reasonable but it is not proved that it is really better than others. Some artefacts are removed - therefore some others (not detected or not presented) are introduced.

The section describing the *impossibility* of comparing bias correction methods is interesting, but at the same time the example is discouraging. If models for present climate have such biases in predicting the $\Delta$ changes than how can we believe their their $\Delta$ changes for the future are reasonable? I missed more discussion on this problem.

The problem physical consistency of bias correction was discussed by the other referee Uwe Ehret. I am surprised that the authors do not consider the problem of consistency on different spatial and temporal scales. From the title I expected discussions in this direction. Hydrological applications require spatial data, therefore spatial correlations should also be considered. Individual corrections do not mean that the correction is correct over different spatial scales. In a previous paper (Bardossy and Pegram 2012) we investigated the spatial coherence of RCMs and found significant differences between model and observations. This problem becomes more severe if the bias corrected output is applied in hydrology. The same also applies for temporal aggregations. Bias correction on a single scale at a single location is an interesting excercise, but may be biased on other scales.

The choice of 0.1 mm daily precipitation threshold is in my opinion not appropriate. Precipitation amounts between 0 and 1 mm are very inaccurate in measurements. They may even contaminate the estimation of the precipitation distributions. I would suggest to use a mixed approach - fitting only above the 1 mm limit.

The gamma distribution for daily precipitation is a reasonable choice, but not for the extremes. It is anyhow very unusual to use the name return periods for relatively frequent events. Please use another notation

The paper is very difficult to read. In my opinion the methods presented in the paper are described in a difficult to follow manner. I spent a lot of time to understand, and finally found that the methods are not very complicated themselves. I personally do not like equations written in a programming language style. There is a correct mathematical description with equations and not using words like sort etc.

In conclusion I find the paper interesting but not well presented and not addressing important issues. Therefore I suggest a major revision.

András Bárdossy

Bárdossy, A. and G. Pegram, Multiscale spatial recorrelation of RCM precipitation to produce unbiased Climate Change scenarios over large areas and small, *Water Resources Research*, **48**, W09502, doi:10.1029/2011WR011524, 2012

---

## Author Comment (AC2) · 30 Jan 2017

Dear Dr. Bárdossy,

The authors would like to thank you for your valuable comments and suggestions. Below we respond to each point.

Reviewer Comment: This paper treats an important problem - how to bias correct RCM output to be used for possible impact assessment. The authors argue that the frequently preferred quantile mapping is under circumstances leading to unreasonable results - therefore another more robust method is needed. The scaled distribution mapping (SDM) suggested by the authors is a sophisticated version of the classical alteration of sequences by multiplication of precipitation and linear scaling of temperature. The method is reasonable but it is not proved that it is really better than others. Some artefacts are removed - therefore some others (not detected or not presented) are introduced.

Author Response: Following the logic that we have outlined in the paper, our method is proven to be better. To sum up our logic from the paper again: We first illustrate that it is not safe to assume that the error correction functions used in quantile mapping (QM) are stationary. And since the inflation/deflation of the climate change signal can be attributed to this stationarity assumption, we argue that QM lacks justification to alter the raw model projected changes. Furthermore, we propose that until a bias correction method can unequivocally demonstrate that it is beneficial to alter raw model projected changes, better performing methods will more effectively preserve these raw model projected changes after bias correction. This is precisely what we have demonstrated. We show how well different methods preserve the raw model projected changes to the leading three moments of the distributions of temperature and precipitation. However, we acknowledge that it would have been useful for us to provide statistical significance measures for the performance metrics. Using a t-test, we find the improved performance of SDM to be statistically significant for the leading three moments. We even find this to be the case for skewness which visually appears very close, though our sample size is quite large (~294,000 mean absolute error values = 3500 grid cells * 12 months * 7 outlook periods).

Yes, it is clear that QM has statistical artifacts as a result of the stationarity assumption. We have developed a method that is free of the stationarity assumption and thus removes the known statistical artifacts inherent to QM. We do concede that some statistical artifacts of a much smaller magnitude could be introduced in our method when scaling between distributions that are poorly fit to the underlying data (this is discussed in more detail in a later comment). Our revised paper will more clearly state the assumptions related to the fitting of the distributions.

Reviewer Comment: The section describing the impossibility of comparing bias correction methods is interesting, but at the same time the example is discouraging. If models for present climate have such biases in predicting the changes than how can we believe their changes for the future are reasonable? I missed more discussion on this problem.

Author Response: Unfortunately, the reviewer misinterpreted the results of this section. First, consider the following two types of bias: 1) the bias between modeled and observed values in some calibration period (the case in the paper is 1951-1980), and 2) the bias between the climate model projected changes and observed changes between the validation period and the calibration period for the leading moments (the case in the paper uses the calibration period 1951-1980 and validation period 1981-2010, for example observations show 1 degree of mean warming over this time period in a specific grid cell while the model shows 1.5 degree of mean warming over the same time period and the same grid cell). The first type of bias is something we can know, though we cannot be sure how it will change under non-stationary. The second type of bias is something we know in the example split-sample validation test. However, in real applications of bias correcting an actual future period (e.g., 2050-2100), we do not know which regions the model will more closely simulate the observed changes that have yet to happen. We do not know if there will be persistence in the ability of the model to correctly simulate observed changes in climate. Likewise, the literature provides no clear evidence that certain model projections should be weighted more than others. This is related to model selection and is beyond the scope of this paper. It is this second type of unknown bias that influences the performance of a bias correction method when using a split-sample test. As a result, we argue that a different type of test should be used to measure the effectiveness of a bias correction method.

Reviewer Comment: The problem physical consistency of bias correction was discussed by the other referee Uwe Ehret. I am surprised that the authors do not consider the problem of consistency on different spatial and temporal scales. From the title I expected discussions in this direction. Hydrological applications require spatial data, therefore spatial correlations should also be considered. Individual corrections do not mean that the correction is correct over different spatial scales. In a previous paper (Bardossy and Pegram 2012) we investigated the spatial coherence of RCMs and found significant differences between model and observations. This problem becomes more severe if the bias corrected output is applied in hydrology. The same also applies for temporal aggregations. Bias correction on a single scale at a single location is an interesting exercise, but may be biased on other scales.

Author Response: The authors gratefully acknowledge the reviewer for providing the valuable reference to the Bardossy and Pegram (2012) paper. It is true that we have proposed a bias correction method that performs grid cell by grid cell. Some users may choose to trust a model with respect to the spatial patterns of meteorological variables, and as a result, they would not match the modeled correlation matrix to that of the observed correlation matrix. However, since we perceive the goal of bias correction is to properly reflect the statistical properties of observations in the calibration period at a variety of scales, we do believe that prior to bias correction it is a wise choice to recorrelate the data as outlined in the reference. This recommendation and the citation will be added to our paper.

Reviewer Comment: The choice of 0.1 mm daily precipitation threshold is in my opinion not appropriate. Precipitation amounts between 0 and 1 mm are very inaccurate in measurements.

They may even contaminate the estimation of the precipitation distributions. I would suggest to use a mixed approach fitting only above the 1 mm limit.

Author Response: We have used a threshold which is the threshold of non-zero precipitation amounts in the E-OBS data set. Different users have different needs. We have used 0.1 mm as a threshold, but this is not a fixed assumption of the method. We will more clearly state in the paper that an appropriate threshold (e.g., 0.1 mm, 0.5 mm, or 1 mm) can and should be chosen by the user to fit their needs.

Reviewer Comment: The gamma distribution for daily precipitation is a reasonable choice, but not for the extremes. It is anyhow very unusual to use the name return periods for relatively frequent events. Please use another notation.

Author Response: Our primary focus was to present a method that preserved raw model changes across the entire distribution. While the method is succeeding in this respect, the choice of a gamma distribution may not always be appropriate. In particular, for extreme events the scaling between return periods can and will often be different depending on the distribution used (e.g. using a weibull instead of a gamma distribution). We need to more clearly state that we are illustrating the method by using a gamma distribution for precipitation, though a user should use the distribution that is most appropriate for their case.

With respect to the name 'return periods,' our method deals with scaling all parts of the distribution and not only the extremes. Technically, we are talking about and using return periods. However, if this is too confusing for the reader, perhaps we could use 'recurrence interval' instead.

Reviewer Comment: The paper is very difficult to read. In my opinion the methods presented in the paper are described in a difficult to follow manner. I spent a lot of time to understand, and finally found that the methods are not very complicated themselves. I personally do not like equations written in a programming language style. There is a correct mathematical description with equations and not using words like sort etc.

Author Response: We specifically wrote the paper and the equations in a style that attempted to be transparent and makes it as easy as possible for a reader to implement themselves. In the end, our method is simply scaling the observed distribution by the amount by which the future modeled distribution is scaled from the historical modeled distribution. Though this is simple in theory, the application becomes a little less elegant. We have outlined the method in a way that we perceived to be appropriate and reader-friendly. If there are more specific suggestions that the reviewer has to improve the readability of the paper, we would enjoy receiving this feedback.

Reviewer Comment: In conclusion I found the paper interesting but not well presented and not addressing important issues. Therefore I suggest a major revision.

Author Response: We respectfully disagree with the reviewer that our paper is not addressing important issues. The authors took the time to understand and implement the spatial recorrelation method described in Bardossy and Pegram (2012). We did this for each month in a 30 year historical time period using bias corrected model precipitation data in a 10 by 10 grid (seen outlined in Figure 1). Similarly to the paper, we find that the model we used underestimates the average correlation between all grid cells with respect to observations. The correlation matrices of the observations, bias corrected model, and the recorrelated bias corrected modeled are shown in Figure 2. After adjusting the bias corrected modeled values using the recorrelation method outlined in the paper, the correlation matrix aligns with the observed correlation matrix (left and right subplots of Figure 2). The spatially averaged daily amounts of modeled precipitation are underestimated for high values and overestimated for low values (though the mean of all these daily values are the same because here we are using bias corrected data). We found the most extreme case of underestimating high values to be for the month of May. Figure 3 shows the sorted observed values versus the sorted modeled values for the month of May in this 10x10 grid before and after the recorrelation. The left subplot shows the shows a systematic underestimation of the uppermost quintile of the spatially averaged modeled precipitation versus observations prior to recorrelation (underestimation because it is below the black 1-to-1 line, the red line is a least squares fit to the data). The amount of the model underestimation averaged across the quintile is approximately 8%. The right subplot shows that with the adjustment, there is no longer this systematic underestimation for this upper quintile (the red line is now obscured by the black line). To reiterate, we find that by not having the correct spatial scale of precipitation events, which can be characterized by the correlation matrix, we would underestimate the spatially averaged precipitation in the upper quintile by 8%. In our paper, it can be seen in Figure 7d, that QM will overestimate the spatially averaged May precipitation by 9% purely as a statistical artifact due to non-stationarity. The magnitude of the problem addressed by recorrelation and the problem of non-stationarity is on the same order. Our proposed method does not rely on this stationarity assumption, and as a result, we remove the systematic inflation/deflation that is directly the result of QM statistical artifacts.

The authors would like to thank the reviewer again for your thoughtful comments.

Yours sincerely,

Matthew Switanek and coauthors

[Figure]

Figure 1: The subregion of 10 by 10 grid cells that we use to implement the recorrelation methodology.

[Figure]

Figure 2: Correlation matrices shown with the colorbar showing correlation coefficients between 0.2 (blue) and 1.0 (red).

[Figure]

Figure 3: Upper quintile of values prior to and after recorrelation. Black lines are 1-to-1, and red lines are fitted least squares lines.

Bárdossy, A. and G. Pegram, Multiscale spatial recorrelation of RCMprecipitation to produce unbiased Climate Change scenarios over large areas and small, Water Resources Research, 48, W09502, doi:10.1029/2011WR011524, 2012

---

## Author Response (AR1)

Dear Dr. Zehe,

The authors thank you for your comments and helpful suggestions. We respond to each point below.

Let me first of all apologize for the fact that my decision on the further treatment of your very interesting manuscript took a while. This is because the reviews of Uwe Ehret and Andras Bárdossy evaluate your work at two rather different, equally important levels – a) the question whether bias correction is a valid procedure in the context of climate change studies and b) the pros and cons of your without doubt interesting approach.

Although the first issue goes partly beyond the scope of your manuscript, HESS is the right journal to debate it. I very much agree that bias correction is a useful procedure in the context of operational forecasting or merging of rainfall radar and rain gauge data, and in fact application of BM has a long tradition in this context (partly under the name model output statistics MOS). In line with the comments of Uwe Ehret I have great concerns about the use of BM in the context of climate change impact studies. Bias correction is a quick fix, to mask our inability to properly reproduce the recent climatologies of target variables, particularly rainfall. A successful reproduction of recent climatologies is a key benchmark to assess whether our climate models and the underlying theories represent the causal effect relation in our climate systems good enough, to expect reliable extrapolations under non-stationary conditions. Systematic model errors suggest that the representation of these cause effect relations is systematically wrong. The current practice is to bias correct the model results – instead of bias correcting the models. Can we promote climate change impact research without bias correction? Yes we can, it is a matter of communicating the current limits of predictability instead of masking them. So what to conclude from this – certainly not stop the search for improved bias correction schemes. But that those schemes should be introduced and discussed with great care, and that development of bias correction schemes cannot substitute bias correction of the models. Your reply to Uwe Ehrets review reveals that you are very much aware of this problem. I thus kindly ask you to avoid the too much enthusiastic opening statement introducing BM as being "essential" you for instance may use "widely used" instead and to critically reflect on these issues in your revised manuscript.

Author response: We have changed some of the wording in the opening paragraph to reflect this suggestion. Additionally, as we wrote in our reply to Dr. Ehret, we have added a paragraph discussing some of the important issues he raised. This added discussion can be found starting at line 450.

That said, let me come to the second level the pros and cons of your without doubt very interesting approach. I dare say that I highly appreciate that your study addresses the stationarity assumption, a key weakness of many bias corrections schemes. Your results show evidently that the stationarity assumption is in valid and that your approach preserves the change signal of the climate model. Yet Andras Bardossy came up with several crucial points that should be addressed or at least discussed in the revised manuscript. One is certainly the issue of physical consistency among different fields and the univariate treatment, neglecting spatial correlations.

Author response: We now state the issue of physical consistency as the first point in the discussion addressing Dr. Ehret's concerns (line 452). As we say, this issue does require consideration and it indeed does merit more focused studies, though it is beyond the scope of our study. We find the recorrelation method, proposed by Dr. Bárdossy, to be of significant value in the context of applying bias corrected data in impact studies that require spatial data (e.g., hydrological modelling). To reflect this, we have added text to advocate the recorrelation method prior to bias correction (line 221). It should be noted, however, that the act of recorrelating the data changes the spatial fields (not unlike bias correction) and thus degrades thermodynamical consistency. In this process, one does gain modelled spatiotemporal fields that are similar to observed fields, but physical model consistency will diminish to some degree.

The second important point is to exclude precipitation amounts within the error margin from the analysis and the suitability of the gamma distribution for rainfall extremes. I think you should check how the proposed distributions change when changing the threshold to 1mm.

Author response: We have run the different methods again using a threshold of 1.0mm. The results of the 1.0 mm versus 0.1 mm thresholds are shown in the new Figure 13 and discussed starting in line 438. Initially, we wanted to provide a threshold where the bias corrected data could have the broadest possible range of applications. However, we do acknowledge that 0.1 mm is an extremely small precipitation amount and could be inside of the margin of error. This was a very helpful suggestion, and we now show that with this regional climate model the method improves with a larger threshold. Though, the choice of an appropriate threshold will ultimately depend on the user and their study.

Allow to point out that a definition of a return period as inverse of the exceedance probability requires equal temporal spacing of the selected sample members (for instance annual maxima). If this is not the case here, the term return period is problematic.

Author response: We have changed the term from return period to recurrence interval to avoid confusion. To be clear, our proposed method is scaling the likelihood of events (by using recurrence intervals) for the entire distribution, not just for the extremes. That said, we respectfully disagree with the point regarding the return period of events (extreme or otherwise) by using equal temporal spacing. Consider a sample data set with 20 years of daily precipitation and that we intend to use the method of annual maxima. Now, assume that the largest 5 maximum daily values took place in one year (each separated by weeks or months and not subject to dependence). If we were to only take the greatest of these 5 values, then we would be missing the 2$^{nd}$ through the 5$^{th}$ largest values from the time series. This would distort the tail behavior of the distribution and would result in a poor estimate of the likelihood of a return event. This logic is also followed in one of our new citations, Papalexiou et al. (2013), which is a HESS paper that deals with precipitation extremes.

The suitability of the gamma function for extremes should, as you already stated in your reply, be discussed; the best passage is the context of your Figure 4, panels b and e.

Author response: The suitability of the gamma versus other distributions is discussed in added content that can be found starting at line 233.

Overall I return this work to you subject to major revisions. Please make sure that the revised manuscript sticks to our mathematical conventions. In line with Andras Bardossy I think that parts of section 3 and 4 should be reformulated as they are not easy to follow.

Author response: Our equations have been changed to follow with mathematical conventions. Please note, that track changes do not show that the equations were altered even when they were.

Best regards,
Matthew Switanek and coauthors

[revised manuscript text omitted]